# Benefits of Physiotherapy on Urinary Incontinence in High-Performance Female Athletes. Meta-Analysis

**DOI:** 10.3390/jcm9103240

**Published:** 2020-10-10

**Authors:** Alba Sorrigueta-Hernández, Barbara-Yolanda Padilla-Fernandez, Magaly-Teresa Marquez-Sanchez, Maria-Carmen Flores-Fraile, Javier Flores-Fraile, Carlos Moreno-Pascual, Anabel Lorenzo-Gomez, Maria-Begoña Garcia-Cenador, Maria-Fernanda Lorenzo-Gomez

**Affiliations:** 1Section of Urology, Department of Surgery, University of Salamanca, 37007 Salamanca, Spain; albash_9@hotmail.com (A.S.-H.); maria.flores.fraile@usal.es (M.-C.F.-F.); mbgc@usal.es (M.-B.G.-C.); mflorenzogo@yahoo.es (M.-F.L.-G.); 2Department of Physiotherapy, University of Salamanca, 37007 Salamanca, Spain; moreno@usal.es; 3Section of Urology, Department of Surgery, University of La Laguna, 38200 Tenerife, Spain; padillaf83@hotmail.com; 4Multidisciplinary Renal Research Group of the Institute for Biomedical Research of Salamanca (IBSAL), 37007 Salamanca, Spain; magalymarquez77@gmail.com; 5Healthcare Complex of Zamora, 49002 Zamora, Spain; alorgom@hotmail.com; 6Urology Service of the University Hospital of Salamanca, 37007 Salamanca, Spain

**Keywords:** physiotherapy, urinary incontinence, high performance athletes

## Abstract

**Introduction:** High performance female athletes may be a risk group for the development of urinary incontinence due to the imbalance of forces between the abdomen and the pelvis. Pelvic floor physiotherapy may be a useful treatment in these patients. **Objectives:** (1) To identify the scientific evidence for pelvic floor (PF) dysfunctions that are associated with urinary incontinence (UI) in high-performance sportswomen. (2) To determine whether pelvic floor physiotherapy (PT) corrects UI in elite female athletes. **Materials and methods:** Meta-analysis of published scientific evidence. The articles analyzed were found through the following search terms: (A) pelvic floor dysfunction elite female athletes; (B) urinary incontinence elite female athletes; (C) pelvic floor dysfunction elite female athletes physiotherapy; (D) urinary incontinence elite female athletes physiotherapy. **Variables studied:** type of study, number of individuals, age, prevalence of urinary incontinence described in the athletes, type of sport, type of UI, aspect investigated in the articles (prevalence, response to treatment, etiopathogenesis, response to PT treatment, concomitant health conditions or diseases. **Study groups according to the impact of each sport on the PF:** G1: low-impact (noncompetitive sports, golf, swimming, running athletics, throwing athletics); G2: moderate impact (cross-country skiing, field hockey, tennis, badminton, baseball) and G3: high impact (gymnastics, artistic gymnastics, rhythmic gymnastics, ballet, aerobics, jump sports (high, long, triple and pole jump)), judo, soccer, basketball, handball, volleyball). Descriptive analysis, ANOVA and meta-analysis. **Results:** Mean age 22.69 years (SD 2.70, 18.00–29.49), with no difference between athletes and controls. Average number of athletes for each study was 284.38 (SD 373,867, 1–1263). The most frequent type of study was case-control (39.60%), followed by cross-sectional (30.20%). The type of UI was most often unspecified by the study (47.20%), was stress UI (SUI, 24.50%), or was referred to as general UI (18.90%). Studies on prevalence were more frequent (54.70%), followed by etiopathogenesis (28.30%) and, lastly, on treatment (17.00%). In most cases sportswomen did not have any disease or concomitant pathological condition (77.40%). More general UI was found in G1 (36.40%), SUI in G2 (50%) and unspecified UI in G3 (63.64%). In the meta-analysis, elite athletes were found to suffer more UI than the control women. In elite female athletes, in general, physiotherapy contributed to gain in urinary continence more than in control women (risk ratio 0.81, confidence interval 0.78–0.84)). In elite female athletes, former elite female athletes and in pregnant women who regularly engage in aerobic activity, physiotherapy was successful in delivering superior urinary continence compared to the control group. The risk of UI was the same in athletes and in the control group in volleyball female athletes, elite female athletes, cross-country skiers and runners. Treatment with PT was more effective in control women than in gymnastics, basketball, tennis, field hockey, track, swimming, volleyball, softball, golf, soccer and elite female athletes. **Conclusions:** There is pelvic floor dysfunction in high-performance athletes associated with athletic activity and urinary incontinence. Eating disorders, constipation, family history of urinary incontinence, history of urinary tract infections and decreased flexibility of the plantar arch are associated with an increased risk of UI in elite female athletes. Pelvic floor physiotherapy as a treatment for urinary incontinence in elite female athletes, former elite female athletes and pregnant athletes who engage in regular aerobic activity leads to a higher continence gain than that obtained by nonathlete women.

## 1. Introduction

Urinary incontinence (UI) is defined as the involuntary loss of urine [1]. When pressure inside the abdomen increases due to exertion, it is transmitted to the bladder causing the pressure within the bladder to be higher than in the urethra. For proper function of urination and urinary continence, intraurethral pressure must be higher than intravesical pressure both at rest and in activities that require effort [2]. The urethra must be evaluated with physical examination and transvaginal ultrasound [3] to rule out disease such as diverticular or rare neoplasms [4]. 

The aim of UI treatment is to regain urinary continence. Treatment begins with conservative measures and changes to lifestyle, followed by pelvic floor (PF) physiotherapy and pharmacological treatments. If this fails, surgical treatment is considered [5]. PF exercises are a fundamental aspect of vesico-sphincter re-education treatments. They were initially proposed by J.W. Davis, although it was Arnold Kegel who described them in detail in 1948, since when they have been known as Kegel exercises [6]. If not performed regularly, the effects of PF exercises may be lost in 10 to 20 weeks after they are stopped [7]. The decision over which PT technique to use should be based on the cause of the urinary incontinence and on the characteristics of the patient, such as learning ability, motivation, and adherence to treatment. Protocols should be followed which begin with the simplest and safest measures, and progress towards the most aggressive guided by pre-established therapeutic objectives (García-Martín, Del-Olmo-Cañas et al. 2005). Biofeedback (BFB) is a method for training pelvic floor muscle exercises by means of positive reinforcement. Through biofeedback we obtain information on the intensity and duration of pelvic floor muscle contractions [7]. The types of BFB most used on the PF are pressure and electromyographic, guided by visual, auditory or tactile stimuli [6]. Another technique is electrostimulation, which can passively enhance PF musculature through the activation of nerve and muscle fibers by electrical or magnetic stimuli [6]. Ptaszkowskin et al. reported good results of stress or mixed urinary incontinence treatments by pelvic floor muscle stimulation with high-inductive electromagnetic stimulation using surface electromyography [8]. However, such treatments are not specified to have been applied to elite female athletes.

The prognosis of UI depends on the type of UI, the age of the patient, past or concomitant illnesses, pregnancy, childbirth and the nature of frequently-performed activities [9]. High-performance female athletes are an at-risk group for UI. This finding could be due to the imbalance of forces in the abdomen and pelvis, which could lead to early alteration of the physiological urethrovesical angle in addition to leading to mixed urinary incontinence (MUI), primarily stress UI. Use of physiotherapy (PT) could prove beneficial in cases of UI in high-performance athletes, both in neuromuscular and fascial aspects, with repercussions on function and prognosis. Practicing various sports may also be associated with different positions of the pelvis or the cowork of synergistic muscles. This is a well-researched aspect in menopausal women [10,11]. In our research, different types of sports practices were taken into account and an international medical expert in sports medicine (Dr Carlos Moreno) classified the different sports according to their impact on the pelvic floor.

The objective of this study was to identify the scientific evidence of association of pelvic floor dysfunction (PF) and urinary incontinence (UI) in high-performance athletes. In addition to determining if pelvic floor (FP) physiotherapy (PT) corrects UI in elite athletes.

## 2. Materials and Methods

A meta-analysis of published scientific evidence was performed. The flow chart shows how the analyzed articles were selected (Figure 1). The time limits of the search are from 1994 to 2019. The main search was carried out in the PubMed database, where the following searches were made. The first search, pelvic floor dysfunction and elite female athletes returned seven articles. In the second search, urinary incontinence and elite female athletes, sixteen were found. The third search, pelvic floor dysfunction and elite female athletes and physiotherapy, returned zero results, but four articles appeared in Google Scholar. Finally, the fourth search, urinary incontinence and elite female athletes and physiotherapy, returned two results. Since some articles were repeated in these search results, a total of eighteen articles were found for analysis.

In the eighteen articles on studies conducted on female athletes, both observational and intervention studies were included.

In the selected articles, the following sports were found: noncompetitive sports (such as hiking), golf, swimming, track and field athletics, throwing athletics, cross-country skiing, field hockey, tennis, badminton, baseball, artistic gymnastics, rhythmic gymnastics, ballet, aerobics, jumping athletics (high, long, triple and pole jump), judo, soccer, basketball, handball, volleyball, and ex-athletes of various specialties. Each sport was assigned a code from least to greatest impact by an expert in sports medicine (Doctor D. Carlos Moreno Pascual), and the sports were grouped in three categories corresponding to the three study groups. In each article we identified the resulting values of the investigated variables for athletes and controls.

### 2.1. Study Groups

Three study groups were established according to the sports’ impact on the PF:

G1: low impact. This group included the following sports: noncompetitive sports, golf, swimming, running athletics, throwing athletics. Although some studies included running sports as moderate impact, our specialist physician in sports medicine, of international recognition (Dr. Carlos Moreno,) considered that the impact on the pelvic floor when running is low.

G2: moderate impact. This group included the following sports: cross-country skiing, field hockey, tennis, badminton, baseball.

G3: high impact. This group included the following sports: gymnastics, artistic gymnastics, rhythmic gymnastics, ballet, aerobics, jumping athletics (high, long, triple and pole jump), judo, soccer, basketball, handball, volleyball.

The specialty of ex-athletes was taken into account and, in articles which specified high performance, thus medium-high levels where weights are often used during training, these athletes were included in G3.

For the meta-analysis, on the one hand, the type of sport performed by each female athlete included in the articles was taken into account. The type of sport was analyzed as a variable. Each sport analyzed was classified in one of the three groups of impact on the pelvic floor. In each article, the elite female athletes, and the subjects who acted as controls in the comparison, were identified. This allowed the multivariate analysis to be carried out, since many publications included various types of sports, it was necessary to analyze with the utmost rigor, disaggregating each sport, individualizing each elite female athlete and the female controls against which they were compared in each article.

### 2.2. Variables Studied

For each article, we analyzed the scientific studies to which they referred, which, in reviews and meta-analyses, may have been more than one. In each study the cases of both female athletes and female controls were analyzed, if they existed. 

The following variables were identified:(a)Type of study. The following types were distinguished: clinical case, bibliographic review, cross-sectional, cross-sectional plus case-control, meta-analysis, randomized trial.(b)Number of individuals participating in each study.(c)Age.(d)Proportion of urinary incontinence described in athletes.(e)Type of sport.(f)Type of urinary incontinence (UI). The following types were noted: general UI, stress UI (SUI), urge UI (UUI), mixed UI (MUI). For greater rigor in the typification of the class of urinary incontinence registered in the articles, we differentiated between type of incontinence not provided (NP), from that specified as general incontinence: general UI.(g)Aspect under investigation in the article. The following study objects were identified: prevalence, response to treatment and etiopathogenesis.(h)Response to physiotherapy treatment. The following responses to physiotherapy were noted: investigated; not investigated. We assessed whether the response to physiotherapy treatment was superior to no physiotherapy treatment. The following responses to physiotherapy treatment were noted: not expressed; expressed and not superior; expressed and superior.(i)Diseases or concomitant health conditions. The following were identified: No concomitant disease; family history of UI, cold environment, embarrassment about urinary incontinence, pregnancy, lower plantar flexibility, eating disorders, constipation, history of urinary tract infections (UTI), levator ani muscle spasm, multiple conditions (family history of UI, constipation plus history of UTI). Cold environment means that the sport is practiced in a cold environment with a low temperature; it is not a disease, as in the case of no concomitant disease. Therefore, cold environment is just a concomitant condition.

### 2.3. Statistical Analysis

The analysis was performed on the NSSS2006/GESS2007 automatic statistical calculator. Descriptive statistics, ANOVA (with Scheffe’s test for normal samples and Kruskal-Wallis for other distributions), and meta-analysis were used. Statistical significance was accepted for *p* < 0.05.

Stages of the analysis: first the type of study was identified, then, within each study, the number of female athletes in each study, the type of sport, the number of controls for each sport.; then the variables studied were identified in each woman. A third step was to catalog the degree of pelvic floor involvement according to the sport studied (which classified the study groups). A final step was to identify the result of the treatment received. The dependent variables were age, number of athletes, number of controls, treatment received and treatment result. The continuous variables were analyzed with one-way ANOVA and the categorical ones with Chi square. All variables were included in the multivariate analysis of randomized effects.

### 2.4. Ethical Issues

The study protocol with code PI2020 03 454 was approved by the Clinical Research Ethics Committee of the Salamanca University Assistance Complex (Comité Ético de Investigación con Medicamentos del Complejo Asistencial Universitario de Salamanca, Salamanca, 37.007 Spain).

### 2.5. Conflicts of Interest

The authors declare there are no conflicts of interest.

### 2.6. Financing

Funding for the study was provided by the Urological Renal Multidisciplinary Research Group (GRUMUR, Grupo de Investigación Multidisciplinar Urológico Renal) of the Salamanca Biosanitary Research Institute (IBSAL, Instituto de Investigación Biosanitaria de Salamanca, Salamanca 37007 Spain).

## 3. Results

Mean age of women in the studies was 22.69 years, SD 2.70, median 22.00, range 18.00–29.49 years, with no differences between athletes and controls (*p* = 0.9864). The mean number of athletes per study was 284.38, SD 373.867, median 144.00, range 1–1263. Mean number of controls was 85.49, SD 175.23, median 0.00, range 0.00–765.00. There were no differences between G1, G2 and G3 in age, number of athletes studied in the articles, or number of control individuals (Table 1).

The most frequent type of study was case-control (39.60%), followed by cross-sectional (30.20%). In G1, case-control studies and meta-analyses were more frequent, in G2 clinical cases and reviews were more frequent, and in G3 cross-sectional studies and randomized trials were more frequent (*p* = 0.000002) (Table 2).

The type of UI was most often unspecified by the study (47.20%), was SUI (24.50%) or was referred to as general UI (18.90%). In G1 the most frequent object of study was general UI (36.40%), followed by MUI (18.20%); in G2, SUI was more frequent (50.00%) and, in G3, type of UI was most frequently unspecified (64.00%) (*p* = 0.0120). Difficulties were found in determining the type of UI in studies which did not specify type of UI or listed as NP (not provided), and in those which did specify general UI (Table 2).

Most of the studies were on the prevalence (54.70%), followed by etiopathogenesis (28.30%) and finally on treatment (17.00%). There were no differences between G1, G2 and G3 (*p* = 0.9600) (Table 2).

Most of the female athletes had no disease or concomitant pathological condition (77.40%). There were no differences between G1, G2 and G3 (*p* = 0.0540) (Table 2).

### Response to Physiotherapy Treatment: Meta-Analysis

Figure 2 shows the meta-analysis of the response to treatment of UI with physiotherapy in female athletes compared to female controls. A fixed effects relative-risk meta-analysis was performed: in the figure. The studies found on the left suggest that physiotherapy favored the group of athletes with urinary incontinence, while those on the right suggest that physiotherapy favored the control group with urinary incontinence. The total effect was RR = 0.81 with confidence intervals (0.78–0.84), so the favorable effect of physiotherapy on athletes with urinary incontinence was significant. That is, effective physiotherapy led to greater symptomatic improvement in athletes versus controls.

Comparisons were made between athletes with urinary incontinence and the control group, in addition to analyzing the characteristics of each study, the number of events (patients with urinary incontinence) and the control group, and the response to treatment of UI with physiotherapy.

## 4. Discussion

This study showed the relationship of the pelvic floor function in elite sportswomen and its association with urinary incontinence. Likewise, it analyzed the results of pelvic floor physiotherapy treatment of urinary incontinence in elite female athletes of different disciplines, comparing them with the results obtained in female controls.

The various randomized clinical studies returned by the PubMed searches showed the effectiveness of PF rehabilitation in UI [12,13,14]. Symptoms of PF dysfunctions may progress and affect an athlete’s life [13].

Several authors argue that strength training of the pelvic floor muscles is effective in the treatment of SUI in women [15,16,17,18,19,20].

A prevalence of UI of 28 to 29.6% of female athletes was reported compared to 9.8 to 13.4% of nonathletes [21,22]. For many young sportswomen who suffer from UI, discussion of the matter remains a taboo in our current society, so providing appropriate information and education on the subject is essential [21].

Women who engage in high-impact sports should be counselled on the impact of these activities on PF function and secondary pelvic floor dysfunctions [23]. Many of these women use vaginal tampons to prevent leakage during high-impact physical activity [20], or use pads or panty liners due to urine leakage [24].

Athletes were observed to have a larger mean diameter of the puborectalis muscle, a greater descent of the bladder neck and a larger urogenital hiatus area during the Valsalva maneuver compared to nonathlete control women, and these changes could potentially be associated with UI [25].

Ex-athletes reported more UI when competing than at the time of taking the questionnaire, years after leaving their sport. Some studies found no difference in UI prevalence between ex-athletes and nonathletes. This indicates that the practice of sport at an early age is not a risk factor for UI in later life in women. It is possible that the musculature of the PF may regain its function after long periods without engaging in high-impact sports, but this aspect needs further investigation [26].

As for pregnant athletes, physical exercise, outside professional and high-performance fields, may be beneficial in maintaining pelvic floor fitness. Women who exercise regularly in the mid-gestation weeks of pregnancy were found to have more endurance and muscle strength in the PF and 31.2% presented with UI in contrast with 38.4% of nonathlete pregnant women [27]. However, engaging in high-performance sports does not prevent the multiple pathologies triggered by childbirth, such as low back pain, UI, or fecal incontinence, but neither does it aggravate those symptoms [28].

Most studies agreed that constipation, a family history of UI, and a history of UTI in female athletes are factors associated with an increased risk of UI [23]. On the other hand, weather conditions during training do not influence the incidence of UI [29]. There is also no evidence that UI is related to amenorrhea, weight, hormonal therapy or the duration of the athletic activity [30].

Eating disorders in female athletes are a risk factor for UI [22,31]. The prevalence of SUI is 49.5% in athletes with eating disorders, compared to 38.8% in healthy athletes [25].

Decreased foot flexibility has been associated with UI in female athletes, suggesting that the manner in which impact forces are absorbed may affect the etiopathogenesis of UI [32].

Although many articles referred to multiple elite sports, to carry out the meta-analysis the type of sport performed by each female athlete included in the study was taken into account. In each article, the elite female athletes and the subjects who acted as controls in the comparison were identified. This allowed multivariate analysis to be carried out, breaking down each of the sportswomen and control women in each sport discipline in each article. The results of our meta-analysis revealed that, in general, female athletes have a higher risk of developing UI compared to nonathlete control women.

In the meta-analysis, the studies with RR lower than 1 showed that physical therapy contributed to gain in urinary continence in patients with urinary incontinence (in female athletes) compared to the control group. Six studies were identified with this result. In elite female athletes these were case-control studies by Bø in 2001 [31] and 2007 [28], a case-control study by Caylet in 2006 [21] and a case-control study by Almeida in 2016 [23]. In former elite female athletes there was a case-control study by Bø in in 2010 [26], and in pregnant women who regularly engage in aerobic activity there was a case-control study by Bø in 2018 [27]. In all these women, physiotherapy as a treatment for UI produced a benefit on continence gain greater than for the control women. The success of PF physiotherapy in these athletes justifies systematically informing women of this benefit.

Studies with RR equal to 1, or those with RR greater than 1 but confidence intervals including 1, showed that the risk of urinary incontinence, and the results of treatment with physiotherapy, was the same in athletes and the control group. Six studies had this result: female volleyball players in a randomized trial by Ferreira in 2014 [12]; elite female athletes, female volleyball players review by Bø in 2004 [33] and elite female athletes in case-control studies by Carvalhais in 2019 [34] and 2018 [22] and by Kruger in 2007 [25]. A similar result was found elite female athlete cross-country skiers and runners in a transversal study by Poswiata in 2014 [29]. These women were at risk of urinary incontinence compared to nonathletes in the control group and, likewise, showed a response to physiotherapy. These results are consistent with the benefit granted by PF physiotherapy in women with UI in the general population [18].

On the other hand, a series of sports were found, such as gymnastics, basketball, tennis, field hockey, track, swimming, volleyball, softball, golf (transversal study by Nygaard in 1994 [30]); female soccer (clinical case by Louis in 2019 [13]); some cases of female volleyball players (randomized trial by Pires in 2020 [14]); and elite female athletes (transversal studies by Nygaard in 1996 [32]) and Thyssen [24]) and meta-analysis by Cerruto in 2019 [35]) where the risk of urinary incontinence was shown to be higher in athletes than in the control group, and where physiotherapy favored the control group more than the athletes. This finding is especially interesting given the poor prognosis associated with these sports.

The conclusions of the study were based on the rigor of the meta-analysis in which the type of sport in each article was stratified, a significant number of women were investigated and the characteristics of both athletes and the controls were considered.

Our analysis found differences in the response to treatment of urinary incontinence in elite female athletes using pelvic floor physiotherapy compared with control women, according to the sport discipline practiced. This is a novel finding that has not been published so far. A stratification and multivariate analysis method was used that distinguished disciplines in which female athletes responded to treatment better than control women, other disciplines where they responded the same and other disciplines in which they responded worse than control women. This is very novel.

Further research is needed on the effect of PT on the pelvic floor and the development of protocols for the treatment of UI by physiotherapists to improve the quality of life of women who practice elite sports.

One weakness of the study is that the results of the analysis refer only to the results of the articles analyzed, and there could be research not included because it was published in other languages or published with keywords other than those used in the search criteria. The search may have limitations because we did not use MESH terms in PubMed and instead used free text.

In the articles analyzed, pelvic floor biofeedback was included as a pelvic floor physiotherapy treatment. It was also necessary that the studies clearly specified type of pelvic floor physiotherapy treatment (biofeedback, electrostimulation, etc.) used.

In the analysis of the heterogeneity of the studies (Figure 3), it is observed that the points included close together under the angle of the funnel plot are more similar, whereas if they are widely dispersed more heterogeneity occurs. In our case, the heterogeneity from a statistical point of view was due to the fact that the studies varied in the number of female athletes studied, the number of female controls employed and the type of studies included (reviews, a clinical case, cross-sectional studies, cases and controls). The latter factor increases heterogeneity. Figure 3 would, preferentially, show the points within the funnel, indicating studies that are similar and justify significant comparisons.

## 5. Conclusions

There is pelvic floor dysfunction in high-performance athletes associated with athletic activity urinary incontinence.

Eating disorders, constipation, a family history of urinary incontinence, a history of urinary tract infections and decreased flexibility of the plantar arch are associated with an increased risk of urinary incontinence in elite female athletes.

Pelvic floor physiotherapy as a treatment for urinary incontinence in elite female athletes, former elite female athletes and pregnant athletes who engage in regular aerobic activity, leads to a gain in continence which is higher than that obtained by nonathlete women.

## Figures and Tables

**Figure 1 jcm-09-03240-f001:**
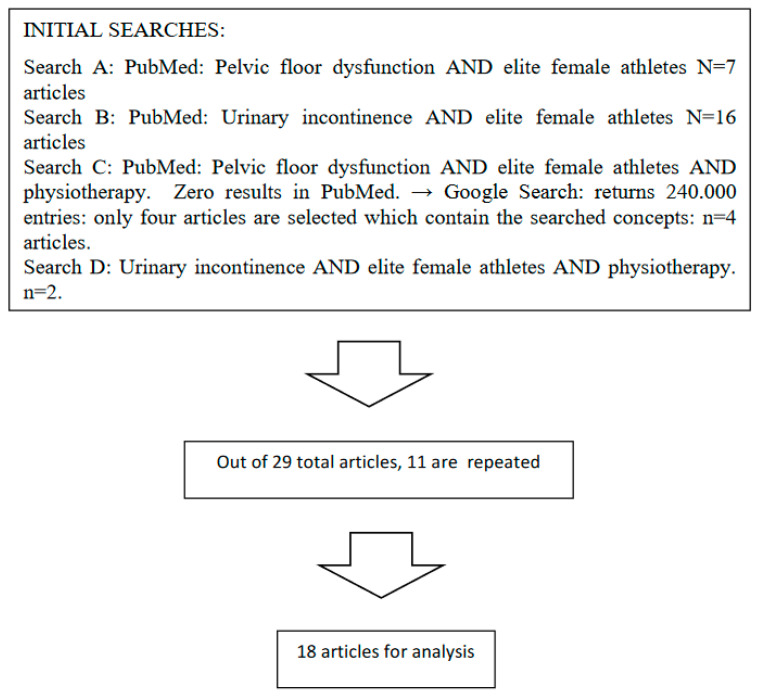
Flow chart of sample selection.

**Figure 2 jcm-09-03240-f002:**
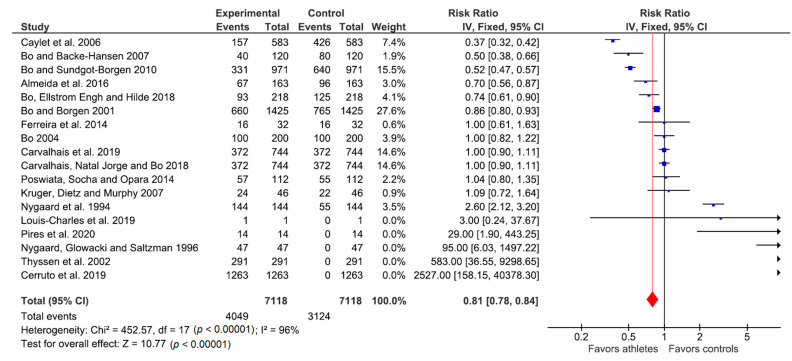
Meta-analysis of studies on the effect of physiotherapy as a treatment for urinary incontinence in female athletes when compared to female controls.

**Figure 3 jcm-09-03240-f003:**
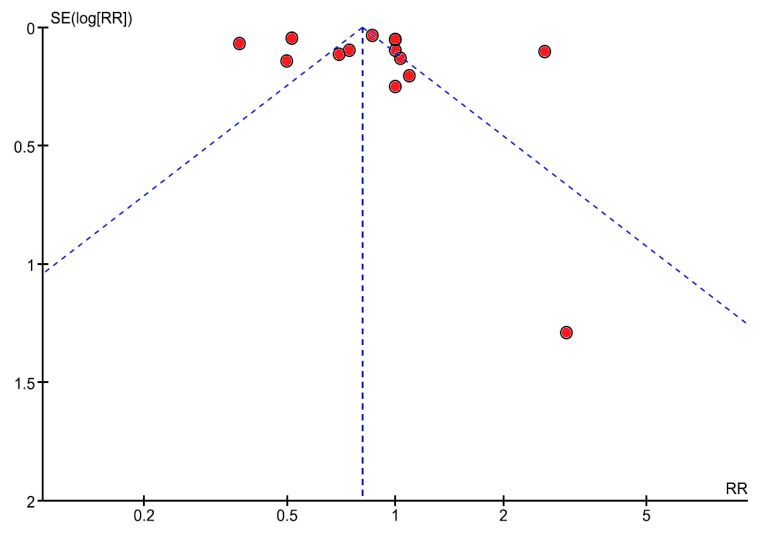
Funnel plot shows the heterogeneity between the studies included in the meta-analysis. Heterogeneity: Chi2 = 452.57, df = 17 (*p* = 0.00001), I2 = 96%.

**Table 1 jcm-09-03240-t001:** Age, number of athletes studied and control women in the studies performed in G1, G2 and G3.

Variable	Group	Mean	SD	Median	Range	*p*
Age	G1	23.37	3.32	22.00	19.00–29.49	0.8000
	G2	22.35	2.94	21.35	19.90–26.61	
	G3	22.18	1.91	22.00	18.00–25.00	
Number of athletes studied	G1	470.91	517.04	150.50	24–1263	0.2470
	G2	146.67	78.83	144.00	57–291	
	G3	153.28	114.43	144.00	1–331	
Number of control women	G1	116.36	195.96	22.00	0.00–765.00	0.3840
	G2	25.83	42.47	0.00	0.00–100.00	
	G3	72.64	174.98	0.00	0.00–640.00	

G1: women engaging in sports with low impact on the pelvic floor. G2: women engaging in sports with moderate impact on the pelvic floor. G3: women engaging in sports with high impact on the pelvic floor.

**Table 2 jcm-09-03240-t002:** Type of studies, type of urinary incontinence investigated, study objective and concomitant diseases or conditions in high-performance sportswomen.

Variable		G1 (*n* = 22 *)		G2 (*n* = 6 *)		G3 (*n* = 25 *)		*p*
		*n*	%	*n*	%	*n*	%	
Type of study	Clinical case	0	0	1	16.70	1	4.00	0.000002
	Review	1	4.5	5	83.30	1	4.00	
	Transversal	5	22.7	0	0	11	44.00	
	Case-control	12	54.5	0	0	9	36.00	
	Meta-analysis	4	18.2	0	0	0	0	
	Randomized trial	0	0	0	0	3	12.00	
Type of UI	General	8	36.40	0	0.00	2	8.00	0.0120
	SUI	3	13.60	3	50.00	7	28.00	
	MUI	4	18.20	1	16.70	0	0.00	
	NP	7	31.80	2	33.30	16	64.00	
Study objective	Prevalence	12	54.50	3	50.00	14	56.00	0.9600
	Treatment	3	13.60	1	16.70	5	20.00	
	Etiopathogenesis	7	31.80	2	33.30	6	24.00	
Concomitant diseases or conditions	None	12	54.50	5	83.30	24	96.00	0.0540
	Cold environment	1	4.50	1	16.70	0	0.0	
	Embarrassment due to UI	1	4.50	0	0.0	0	0.0	
	Pregnancy	3	13.60	0	0.0	0	0.0	
	Less flexibility	2	9.10	0	0.0	0	0.0	
	Eating disorder	1	4.50	0	0.0	0	0.0	
	Muscle spasms	0	0.0	0	0.0	1	4.00	
	Multiples diagnoses	2	100.0	0	0.0	0	0.0	

G1: women engaging in sports with low impact on the pelvic floor. G2: women engaging in sports with moderate impact on the pelvic floor. G3: women engaging in sports with high impact on the pelvic floor. UI: urinary incontinence. SUI: stress urinary incontinence. MUI: mixed urinary incontinence. NP: not provided. * Number of studies which included sporting disciplines categorised as having low, moderate and high impact on the pelvic floor. Some studies included multiple disciplines.

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
