# Peer review of "Benefits of Physiotherapy on Urinary Incontinence in High-Performance Female Athletes. Meta-Analysis"

_jcm, 2020, doi:10.3390/jcm9103240_

Round 1

Reviewer 1 Report

The authors present their results from a meta-analysis of literature surrounding the effect of athletics on pelvic floor dysfunction/urinary incontinence and the role of pelvic floor physiotherapy. This topic is important to many patients and research participation and is a valuable addition to the field. There are a few opportunities for clarification throughout the manuscript, which I have identified below.

Abstract: The abstract is well-written and concise.

Line 41 – The use of the word “protective” is misleading in this context. “Protective” implies PT will prevent the development of UI, although the objectives specifically state that PT was analyzed as a treatment. This needs to be clarified and consistent. This term is also used in the Results, line 226.

Line 48 – The term, “causes”, is too strong. This analysis simply demonstrates an association between athletic activity and UI. No data in this manuscript is sufficient to imply causation. Please change this term here as well as in line 265, and line 209.

Introduction:

Lines 58-71 – Consider combining these into a single paragraph so that paragraphs are composed of more than a single sentence.

Lines 92-94 – These sound like the objectives of the manuscript and are redundant with the following lines (95-97). Lines 92-94 can be deleted or restated as the objectives with the elimination of lines 95-97.

Methods:

Lines 100-101 – This statement is repeated in the following statement. It can be removed.

Study groups: please expand on the decision to include running sports as “low impact”. Many experts (and many studies) classify running as moderate impact at least.

Study groups – how were these groups maintained (or not) in the meta-analysis? Were all athletes combined or were the groups maintained? This needs to be clarified in order to be able to interpret the results of the meta-analysis. It should also be included in the Discussion section as it likely impacts the findings from the meta-analysis.

Line 160 – please expand on the meaning of “cold environment” in the concomitant health conditions. Does this refer to practicing in cold weather (lines 279-280)? If so, this is not really a concomitant condition. Regardless, there needs to be additional clarity.

Results:

Lines 197-198 – why did the authors choose to distinguish “Not specified” from “General UI”? It seems these could be combined. If there was rationale for the distinction, this should be explained in the Methods.

Lines 214-217 – This is confusing, please rephrase. It seems PT leads to greater symptomatic improvement in athletes vs controls, which would be a much clearer way to interpret the findings from the meta-analysis.

Line 226 – Again, I suspect “protective factor” is used incorrectly here.

Lines 229 – 248 – This is redundant with the information presented in Figure 2 and should be removed. Instead, it would be better if this information were summarized and interpreted in the context of the findings of the meta-analysis as part of the discussion section. Given the variability in included athlete types, what do the authors conclude from the findings of the meta-analysis?

Discussion:

Lines 250-252 – The first statement of the Discussion should summarize the findings of the study.

Lines 253-254 – Data besides that published by Bo and colleagues supports this statement

Line 260 – Again, this statement does not appear to be supported by the data in this meta-analysis as the stated objective was to investigate the role of PT as a treatment, not a preventative agent. If the included studies investigated this as a preventative agent, this should be clarified, explained, and discussed throughout the text.

Line 265 – Again, “cause” is too strong. This study does not demonstrate causation, just an association.

Lines 297-303 – This statement directly contradicts the findings presented in this meta-analysis. There should be greater expansion and discussion of these differential findings – why do the authors suspect their findings differed from what has been presented in the literature? What factors differentiate this study from the one they cite?

There is no statement of strengths/weaknesses. In particular, this was not a systematic review of the literature, which should be clearly stated as a weakness as the findings from the meta-analysis are only as strong as the included study. The absence of a systematic literature review risks missing key manuscripts/data.

Author Response

I PUT IN RED THE HIGHLIGHTED CORRECTIONS ACCORDING TO THE COMMENTS OF REVIEWER 1:

I PUT IN RED THE HIGHLIGHTED CORRECTIONS  ACCORDING TO THE COMMENTS OF REVIEWER 1.

The authors present their results from a meta-analysis of literature surrounding the effect of athletics on pelvic floor dysfunction/urinary incontinence and the role of pelvic floor physiotherapy. This topic is important to many patients and research participation and is a valuable addition to the field. There are a few opportunities for clarification throughout the manuscript, which I have identified below.

Abstract: The abstract is well-written and concise.

Line 41 – The use of the word “protective” is misleading in this context. “Protective” implies PT will prevent the development of UI, although the objectives specifically state that PT was analyzed as a treatment. This needs to be clarified and consistent. This term is also used in the Results, line 226.

The reviewer is right: we change: “it was protective” for “it contributed to gain in urinary continence”

Line 48 – The term, “causes”, is too strong. This analysis simply demonstrates an association between athletic activity and UI. No data in this manuscript is sufficient to imply causation. Please change this term here as well as in line 265, and line 209.

The reviewer is right: we change: "cause" to "is associated"

Introduction:

Lines 58-71 – Consider combining these into a single paragraph so that paragraphs are composed of more than a single sentence.

We reduce lines 58 to 71 to a single paragraph, eliminating two sentences and summarizing.

Lines 92-94 – These sound like the objectives of the manuscript and are redundant with the following lines (95-97). Lines 92-94 can be deleted or restated as the objectives with the elimination of lines 95-97.The reviewer is right: we removed lines 92-94.

Methods:

Lines 100-101 – This statement is repeated in the following statement. It can be removed.

The reviewer is right: we removed lines 100-101.

Study groups: please expand on the decision to include running sports as “low impact”. Many experts (and many studies) classify running as moderate impact at least.

This explanatory phrase is added:Although some studies include running sports as moderate impact, our Specialist Physician in Sports Medicine, of international recognition (Dr. Carlos Moreno) considered that the impact on the pelvic floor when running is low.

Study groups – how were these groups maintained (or not) in the meta-analysis? Were all athletes combined or were the groups maintained? This needs to be clarified in order to be able to interpret the results of the meta-analysis. It should also be included in the Discussion section as it likely impacts the findings from the meta-analysis.

This explanatory phrase is added in the Methods:For the meta-analysis, on the one hand, the type of sport performed by each female athlete included in the articles was taken into account. The type of sport was analyzed as a variable. Each sport analyzed was classified in one of the three groups of impact on the pelvic floor. In each article, the elite female athletes and the subjects who acted as controls in the comparison were identified. This allowed the multivariate analysis to be carried out, since many publications included various types of sports and they had to be analyzed with the utmost rigor, disaggregating each and every one of the sports and female controls to which it referred to in each article.

In the Discussion, at the level of line 286, we add this explanation:

Although many articles refer to multiple elite sports, to carry out the meta-analysis, on the one hand, the type of sport performed by each female athlete included in the study was taken into account. In each article, the elite female athletes and the subjects who acted as controls in the comparison were identified. This allowed the multivariate analysis to be carried out, breaking down in each article each and every one of the sports and control women to which he referred. Line 160 – please expand on the meaning of “cold environment” in the concomitant health conditions. Does this refer to practicing in cold weather (lines 279-280)? If so, this is not really a concomitant condition. Regardless, there needs to be additional clarity.

ANSWER:Indeed "cold environment" means that the sport is practiced in a cold environment, with a low temperature. It is not a disease, just as "no concomitant disease". Therefore, “cold environment” is just a concomitant condition.We add this explanation. 

Results:

Lines 197-198 – why did the authors choose to distinguish “Not specified” from “General UI”? It seems these could be combined. If there was rationale for the distinction, this should be explained in the Methods.This sentence is added in Methods at the level of lines 151-152:

For greater rigor in the typification of the class of urinary incontinence registered in the articles, we have differentiated between “type of incontinence not provided: NP, from that specified as general incontinence: general UI.

Lines 214-217 – This is confusing, please rephrase. It seems PT leads to greater symptomatic improvement in athletes vs controls, which would be a much clearer way to interpret the findings from the meta-analysis.

Indeed, the reviewer is right: we added this explanatory and clarifying sentence at the level of line 219. That is, effectively physiotherapy leads to greater symptomatic improvement in athletes versus controls.

Line 226 – Again, I suspect “protective factor” is used incorrectly here.

ANSWER:

We have changed it to: “physical therapy contributed to gain in urinary continencein patients with urinary incontinence”

Lines 229 – 248 – This is redundant with the information presented in Figure 2 and should be removed. Instead, it would be better if this information were summarized and interpreted in the context of the findings of the meta-analysis as part of the discussion section.

Answer: The authors pass the information on lines 229-248 to the discussion section, in summary form. Go to line 288.

Given the variability in included athlete types, what do the authors conclude from the findings of the meta-analysis?

ANSWER:

We add this phrase in the discussion:

Due to the rigor of the meta-analysis, where in each article the type of sport analyzed has been stratified, the number of women investigated and what characteristics both the athletes and the controls against which they have been compared had, the results of the meta-analysis lead to the conclusions.

Discussion:

Lines 250-252 – The first statement of the Discussion should summarize the findings of the study.

ANSWER:We put as the first sentence of the discussion:

This study shows the relationship of the pelvic floor function in elite sportswomen and its association with urinary incontinence. Likewise, it analyzes the results of pelvic floor physiotherapy treatment of urinary incontinence in elite female athletes of different disciplines, comparing them with the results obtained in female controls.

Lines 253-254 – Data besides that published by Bo and colleagues supports this statement

ANSWER:

We change this sentence to:

Several authors argue that strength training of the pelvic floor muscles is effective in the treatment of SUI in women(Bø 2004, Neumann, Grimmer et al. 2006, Lorenzo-Gomez, Silva-Abuin et al. 2008, Oliveira, Ferreira et al. 2017, Dumoulin, Cacciari et al. 2018, Campbell, Batt et al. 2020).

++++++++++++++++++++++++++++++++++++

Line 260 – Again, this statement does not appear to be supported by the data in this meta-analysis as the stated objective was to investigate the role of PT as a treatment, not a preventative agent. If the included studies investigated this as a preventative agent, this should be clarified, explained, and discussed throughout the text.The reviewer is right: where it says: strategies for preventing, has to put: need for treatment

Line 265 – Again, “cause” is too strong. This study does not demonstrate causation, just an association.

ANSWER:

WE HAVE CHANGE TO:  associated with

Lines 297-303 – This statement directly contradicts the findings presented in this meta-analysis. There should be greater expansion and discussion of these differential findings – why do the authors suspect their findings differed from what has been presented in the literature? What factors differentiate this study from the one they cite?

Answer:

We add these comments in the discussion, after lines 297-303:

Our analysis finds differences in the response to treatment of urinary incontinence in elite female athletes, using pelvic floor physiotherapy, when compared with control women, according to the sport discipline practiced. This is a novel finding that has not been published so far. A stratification and multivariate analysis method has been used that distinguishes disciplines where female athletes respond to treatment better than control women, other disciplines where they respond the same, and other disciplines where they respond worse than control women. Which is very novel.

There is no statement of strengths/weaknesses. In particular, this was not a systematic review of the literature, which should be clearly stated as a weakness as the findings from the meta-analysis are only as strong as the included study. The absence of a systematic literature review risks missing key manuscripts/data.

Answer:

We add these comments at the end of the discussion:

One weakness of this study is that the results of the analysis refer only to the results of the articles analyzed, and there could be research not included, because they were published in other languages ​​or were published with keywords other than those used in the search criteria.

Reviewer 2 Report

The reviewis well done and written.The clinical and pathological messages are clear and based on data.

I suggest minor modifications:

  1. Authors should declare in the methods section: The time limits of the search: state from … to..
  2. The search may have limitation because they did not use MESH terms in PubMed, instead “free text”
  3. It is not clearly state if Biofeedback, a method for training pelvic floor muscle, is included in Pelvic floor physiotherapy or excluded or considered separately.
  4. I suggest to add in the introduction section.: the urethra is evaluated with physical examination and transvaginal ultrasound [galosi etl al ] to role out disease such as diverticular or rare neoplasms [Dell’Atti ].
  • Galosi AB, Dell’Atti L. Ultrasound Study of the Urethra. Chapter 18 Pag 211-226. In Book: Atlas of Ultrasonography in Urology, Andrology, and Nephrology. P. Martino, A.B. Galosi (eds.) Springer International Publishing Switzerland 2017.
  • Dell'Atti L, Galosi AB. Female Urethra Adenocarcinoma. Clin Genitourin Cancer. 2018;16(2):e263-267.

Author Response

The answers and corrections suggested by reviewer 2 are expressed in green:

The reviewis well done and written.The clinical and pathological messages are clear and based on data.

I suggest minor modifications:

1.-Authors should declare in the methods section: The time limits of the search: state from … to..

ANSWER:

It is added in Methods:

The time limits of the search state from 1994 to 2019.

2.-The search may have limitation because they did not use MESH terms in PubMed, instead “free text”.

ANSWER:

It is added in Discussion:

The search may have limitation because we did not use MESH terms in PubMed, instead  we used “free text”.

3.-It is not clearly state if Biofeedback, a method for training pelvic floor muscle, is included in Pelvic floor physiotherapy or excluded or considered separately.

ANSWER:

It is added in Discussion:

In the articles analyzed, pelvic floor biofeedback was included as a pelvic floor physiotherapy treatment. It is also necessary that the studies clearly specify what type of pelvic floor physiotherapy treatment (biofeedback, electrostimulation, etc.) has been used.

4.-I suggest to add in the introduction section.:the urethra is evaluated with physical examination and transvaginal ultrasound [galosi etl al ] to role out disease such as diverticular or rare neoplasms [Dell’Atti ].

  • Galosi AB, Dell’Atti L. Ultrasound Study of the Urethra. Chapter 18 Pag 211-226. In Book: Atlas of Ultrasonography in Urology, Andrology, and Nephrology. P. Martino, A.B. Galosi (eds.) Springer International Publishing Switzerland 2017.
  • Dell'Atti L, Galosi AB. Female Urethra Adenocarcinoma. Clin Genitourin Cancer. 2018;16(2):e263-267.

ANSWER:

We add this in the introduction.

Reviewer 3 Report

The manuscript presents an important and interesting topic. Comments for the authors are presented below:

The introduction should be redrafted. It should contain three / four paragraphs (not 13 paragraphs) related thematically (e.g. introductory information about UI, factors influencing UI, briefly current methods of treatment, the purpose of the work).

In the introduction, it would also be worth paying attention to other, new methods of UI treatment:
Ptaszkowski K, Malkiewicz B, Zdrojowy R, Ptaszkowska L, Paprocka-Borowicz M. Assessment of the Short-Term Effects after High-Inductive Electromagnetic Stimulation of Pelvic Floor Muscles: A Randomized, Sham-Controlled Study. J Clin Med. 2020 Mar 23;9(3):874. doi: 10.3390/jcm9030874. PMID: 32210031; PMCID: PMC7141507.

Practicing various sports may also be associated with a different position of the pelvis or the co-work of synergistic muscles, which is worth mentioning.Ptaszkowski K, Zdrojowy R, Slupska L, Bartnicki J, Dembowski J, Halski T, Paprocka-Borowicz M. Assessment of bioelectrical activity of pelvic floor muscles depending on the orientation of the pelvis in menopausal women with symptoms of stress urinary incontinence: continued observational study. Eur J Phys Rehabil Med. 2017 Aug;53(4):564-574. doi: 10.23736/S1973-9087.17.04475-6. Epub 2017 Jan 30. PMID: 28145398.

Ptaszkowski K, Zdrojowy R, Ptaszkowska L, Bartnicki J, Taradaj J, Paprocka-Borowicz M. Electromyographic evaluation of synergist muscles of the pelvic floor muscle depending on the pelvis setting in menopausal women: A prospective observational study. Gait Posture. 2019 Jun;71:170-176. doi: 10.1016/j.gaitpost.2019.04.024. Epub 2019 Apr 25. PMID: 31075659.

Statistical analysis section. Describe exactly how the stages of the analysis were carried out, taking into account:
describe which dependent variables or summary measures are allowed (Differences, Means, Hedges')
Indicate the selection of a meta-analysis model, e.g. fixed effect or random effects meta-analysis.
How assumptions (e.g., heterogeneities) were checkedResults.
Please add the distribution of effect sizes, visualized with a funnel plot.

Author Response

The answers and corrections suggested by reviewer 3 are expressed in blue:The manuscript presents an important and interesting topic. Comments for the authors are presented below:

1.-The introduction should be redrafted. It should contain three / four paragraphs (not 13 paragraphs) related thematically (e.g. introductory information about UI, factors influencing UI, briefly current methods of treatment, the purpose of the work).

ANSWER:

The contents have been reordered and the introduction reduced to 4 paragraphs.

2.-In the introduction, it would also be worth paying attention to other, new methods of UI treatment:
Ptaszkowski K, Malkiewicz B, Zdrojowy R, Ptaszkowska L, Paprocka-Borowicz M. Assessment of the Short-Term Effects after High-Inductive Electromagnetic Stimulation of Pelvic Floor Muscles: A Randomized, Sham-Controlled Study. J Clin Med. 2020 Mar 23;9(3):874. doi: 10.3390/jcm9030874. PMID: 32210031; PMCID: PMC7141507.

ANSWER:

These sentences have been added to the introduction: Ptaszkowskin et al. reported good results of stress or mixed urinary incontinence treatment by pelvic floor muscles stimulation with high-inductive electromagnetic stimulation by using surface electromyography (Ptaszkowski K, Malkiewicz B, Zdrojowy R, 2020). However, such treatment is not specified to have been applied to elite female athletes.

3.Practicing various sports may also be associated with a different position of the pelvis or the co-work of synergistic muscles, which is worth mentioning.Ptaszkowski K, Zdrojowy R, Slupska L, Bartnicki J, Dembowski J, Halski T, Paprocka-Borowicz M. Assessment of bioelectrical activity of pelvic floor muscles depending on the orientation of the pelvis in menopausal women with symptoms of stress urinary incontinence: continued observational study. Eur J Phys Rehabil Med. 2017 Aug;53(4):564-574. doi: 10.23736/S1973-9087.17.04475-6. Epub 2017 Jan 30. PMID: 28145398.

Ptaszkowski K, Zdrojowy R, Ptaszkowska L, Bartnicki J, Taradaj J, Paprocka-Borowicz M. Electromyographic evaluation of synergist muscles of the pelvic floor muscle depending on the pelvis setting in menopausal women: A prospective observational study. Gait Posture. 2019 Jun;71:170-176. doi: 10.1016/j.gaitpost.2019.04.024. Epub 2019 Apr 25. PMID: 31075659.

ANSWER:

These sentences have been added to the introduction:

Practicing various sports may also be associated with a different position of the pelvis or the co-work of synergistic muscles.
This is a well-researched aspect in menopausal women (Ptaszkowski K, Zdrojowy R, Slupska L, 2017; Ptaszkowski K, Zdrojowy R, Ptaszkowska L, 2019. In our research, the different types of sports practices were taken into account, and an international medical expert in Sports Medicine (Dr Carlos Moreno) classified the different sports according to their impact on the pelvic floor.

4.-Statistical analysis section. Describe exactly how the stages of the analysis were carried out, taking into account:
describe which dependent variables or summary measures are allowed (Differences, Means, Hedges')
Indicate the selection of a meta-analysis model, e.g. fixed effect or random effects meta-analysis.
How assumptions (e.g., heterogeneities) were checked

ANSWER:We add this sentence in Methods:
Stages of the analysis: First the type of study was identified, then, within each study, the number of female athletes in each study, the type of sport, the number of controls for each sport. Then the variables studied were identified in each woman. A third step was to catalog the degree of pelvic floor involvement according to the sport studied (which classifies the study groups). A final step is to identify the result of the treatment received. The dependent variables were age, number of athletes, number of controls, treatment received, treatment result. The continuous variables were analyzed with one-way ANOVA, the categorical ones with Chi square. All variables were included in the multivariate analysis of randomized effects.We add this sentence in Discussion:In the analysis of the heterogeneity of the studies (Figure 3), it is observed that the points that are included under the angle of the funnel plot and are together are more similar, and if they are widely dispersed there is more heterogeneity. In our case, the heterogeneity from the statistical point of view is due to the fact that the studies vary in the number of female athletes studied, the number of female controls employed, the type of studies included (reviews, a clinical case, cross-sectional studies, cases and controls). The latter is the factor that enhances heterogeneity. Figure 3 preferably shows the points within the funnel, thus, they are studies that are similar and that justify significant comparisons.5.-Results.
Please add the distribution of effect sizes, visualized with a funnel plot.

ANSWER:

We add in Results the figure 3:

Figure 3.-Funnel plot shows the heterogeneity between the studies included in the meta-analysis. Heterogeneity: Chi2 = 452.57, df = 17 (p = 0.00001), I2 = 96%.

Round 2

Reviewer 1 Report

The modifications have strengthened this manuscript. I have a few remaining comments.

Line 242 - "effectively" should be replaced with "effective"

Line 289 - rephrase to "need for treatment of..."

Line 320 - the use of "he" in this sentence is unclear. Please rephrase.

Line 330 - replace "this" with "these"

Lines 350-353 - This is a run-on-sentence that is difficult to interpret. Recommend rephrasing, potentially with more, shorter sentences to improve clarity.

Author Response

ANSWER TO REVIEWER:

1.-Line 242: “effectively” should be replaced with “effective”:

ANSWER:

We replace effectively  with effective.

2.-Line 289: rephrase to “need for treatment of …”

ANSWER:

We rephrase: we delete “need for treatment of”, therefore the phrase will be:

“Women who engage in high-impact sports should be counselled on the impact of these activities on PF function and secondary pelvic floor dysfunctions”

3.-Line 320: the use of “he” in this sentence is unclear. Please rephrase.

ANSWER:

it's a mistake. where says:

“breaking down in each article each and every one of the sports and control women to which hereferred

should say:

“breaking down each and every one of the sportswomen and control women in each sport discipline in each article”.

4.-Line 330: replace “this” with “these”.

ANSWER: we have replaced it.

5.-Lines 350-353: This is a run-on-sentence that is difficult to interpret. Recommend rephrasing, potentialy with more, shorter sentences to improve clarity.

ANSWER:

We substitute:Due to the rigor of the meta-analysis, where in each article the type of sport analyzed has been stratified, the number of women investigated and what characteristics both the athletes and the controls against which they have been compared had, the results of the meta-analysis lead to the conclusions.

By:

To obtain the conclusions, the key has been rigor in the meta-analysis.

For the analysis, in each article, the results were stratified by the type of sport analyzed, the number of women investigated and the characteristics of the female athletes and the control women against whom they were compared.

Reviewer 3 Report

Thank you for all corrections. 

Author Response

Good Morning Many thanks for your comments and appreciations.